Identification of potential functional peptides involved in demyelinating injury in the central nervous system

Dong Xiaohua 1
Sun Shuchen 1
Li Jie 1
Shen Sen 1
Chen Wanting 2
http://orcid.org/0000-0001-9175-4657 Li Tongqi 3 sanmu27@yeah.net
Li Xinyuan 1 lxy2186@shtrhospital.com
1 Department of Neurosurgery, Tongren Hospital, Shanghai Jiao Tong University School of Medicine , Shanghai , China
2 Jiangnan University , Wuxi , China
3 Department of Ophthalmology, Shanghai General Hospital, Shanghai Jiao Tong University School of Medicine , Shanghai , China
Uversky Vladimir
Electronic publication date: 2023 Aug 21
Publication date: 2023
Volume: 11
Electronic Location ID: e15846
Received 2023 Apr 19; Accepted 2023 Jul 14
Copyright: © 2023 Dong et al.
Copyright year: 2023
Copyright holder: Dong et al.
License: This is an open access article distributed under the terms of the Creative Commons Attribution License, which permits unrestricted use, distribution, reproduction and adaptation in any medium and for any purpose provided that it is properly attributed. For attribution, the original author(s), title, publication source (PeerJ) and either DOI or URL of the article must be cited.
License URL: https://creativecommons.org/licenses/by/4.0/

Keywords: Peptides, Multiple sclerosis, Demyelinating injury, Central nervous system, MBP

Funding: Natural Science Foundation of Shanghai 22ZR1457100 Famous doctor’s workshop in Changning District of Shanghai MYGZS007 This work was supported by the Natural Science Foundation of Shanghai under Grant No. 22ZR1457100, and the Famous doctor’s workshop in Changning District of Shanghai (MYGZS007). The funders had no role in study design, data collection and analysis, decision to publish, or preparation of the manuscript.

==============================
Multiple sclerosis (MS) is a chronic inflammatory neurologic disease characterized by the demyelinating injury of the central nervous system (CNS). It was reported that the mutant peptide came from myelin proteolipid protein (PLP) and myelin basic protein (MBP) might play a critical role in immunotherapy function of MS. However, endogenous peptides in demyelinating brain tissue of MS and their role in the pathologic process of MS have not been revealed. Here, we performed peptidomic analysis of freshly isolated corpus callosum (CC) from the brains of CPZ-treated mice and normal diet controls of male C57BL/6 mice by LC-MS/MS. Identified a total of 217 peptides were expressed at different levels in MS mice model compared with controls. By performed GO (Gene Ontology) and KEGG (Kyoto Encyclopedia of Genes and Genomes) analysis, we found that the precursor protein of these differently expressed peptides (DEPs) were associated with myelin sheath and oxidative phosphorylation. Our study is the first brain peptidomic of MS mice model, revealing the distinct features of DEPs in demyelination brain tissue. These DPEs may provide further insight into the pathogenesis and complexity of MS, which would facilitate the discovery of the potential novel and effective strategy for the treatment of MS.

Introduction

Multiple sclerosis (MS) is an immune-mediated disease characterized by demyelination, inflammatory, and axonal injury in the central nervous system (CNS) (Yamout & Alroughani, 2018). The average annual incidence rate of MS was 33/100,000, and the incidence and prevalence of MS have increased in both in developed and developing countries (Browne et al., 2014; Oh, Vidal-Jordana & Montalban, 2018; Talebi et al., 2021). MS usually affects young adults and leads to neurological disability (Salles et al., 2022), resulting in the social and economic burden of the family. It was reported that genetic, environmental, and epigenetic factors involved in the pathological of MS (Kurtzke, 2013; Oh, Vidal-Jordana & Montalban, 2018; Thompson et al., 2018), however, the ultimate cause of MS is unknown. Current treatment strategies for multiple sclerosis focus on managing acute attacks with disease-modifying therapies, which are primarily based on targeted inflammatory therapy for these patients. However, progressive disability and early mortality of MS remain a concern (Gholamzad et al., 2019). Therefore, there is an urgent need to focus on the pathological process of multiple sclerosis and develop new treatments.

The myelin of the CNS plays a vital role in axon function by providing metabolic support to the axon and allowing action potential to be rapidly transmitted along axon (Smith, Blakemore & McDonald, 1979; Nave, 2010; Franklin & Ffrench-Constant, 2017). Demyelination of axons (loss of myelin sheath) in CNS is a pathological hallmark of MS. Myelin basic protein (MBP) and myelin-associated glycoprotein (MAG) are important for oligodendrocyte-mediated myelin sheath maturation (Uschkureit et al., 2000; Han et al., 2020b). Besides, the immune cell infiltration (CD4+ T cells, CD8+ T cells, macrophages and Th17 cells) and destruction of myelin sheath is involved in the pathophysiology of MS (Katsara et al., 2008; Dargahi et al., 2017; Katsara & Apostolopoulos, 2018). Activating the remyelination of oligodendrocytes (OLs) in the CNS is necessary for the rehabilitation of MS.

Peptide was considered to be a better candidate for treating human diseases due to its specific biochemical properties (e.g., better permeability) and therapeutic properties (Wang et al., 2022; Tramutola et al., 2023). Peptides are fragments of proteins that produced by the protein hydrolysis and carry out important biological functions (Apostolopoulos et al., 2021). Multiple evidence suggests that peptides exhibit a powerful role in modulating neuronal activity. For example, a short peptide-based inhibitor comprised of D-amino acids could inhibits fibril formation by interacting with amyloid β aggregation in Alzheimer’s disease (Baig et al., 2018; Horsley et al., 2020). Naturido is an anti-aging cyclic peptide that promotes astrocyte proliferation and activates VGF and NGF signaling (Ishiguro et al., 2021). Peptide (TQKKVIFC) which derived from beef myofibrillar protein could prevent neuronal cell death (Lee & Hur, 2019). Peptides act as signal transducers, interfering with interactions between proteins that could be used for further treatment.

In MS, it was reported that the cyclic peptide MBP87-99 could alter immune responses in MS mouse model and might be a therapeutic approach for MS (Katsara et al., 2008, 2008, 2009, 2014). In addition, modified PLP139-151 peptide has potential immunotherapy of MS (Katsara et al., 2014). Although it has been identified MBP and PLP peptide epitope may have a potential role in MS therapy. However, whether more endogenous peptides participate in the pathological process of MS is undetermined.

In this study, we established the CPZ (bis-cyclohexanone-oxaldihydrazone)-induced mouse model which mimic the human MS disease, and identified a total of 217 differentially expressed peptides (DEPs) in the corpus callosum (CC) of central nervous system by LC-MS/MS. GO and KEGG analysis reveals that DEPs involved in the myelin sheath progress and oxidative phosphorylation pathway, indicated these DEPs might play an essential role in the pathological of MS.

Materials and Methods

Animals

Six-week-old male C57BL/6 mice were purchased from SPF (Beijing) Biotechnology Co., LTD (Beijing, China) and raised and maintained for 2 weeks on the ventilated conditions (22 °C, 40–60% humidity, 12 h light/dark) before experiment beginning. Eight-week-old mice were randomization divided into two groups (n = 6 mice/group), the experimental animals were numbered and assigned into the control group and the experimental group through random selection of numbers. The control group fed with normal rodent food, Cuprizone (CPZ) groups were fed with 0.2% CPZ-containing (Sigma, St. Louis, MO, USA) diet for 8 weeks to induce demyelination. Mice were weighed on a daily basis. Animals were sacrificed by dislocation of cervical vertebra for the experiments right after the end of 8 weeks-fed. In this study, six mice were used in each treatment group. Three were sacrificed for histological and IF study, while the other three were sacrificed for LC-MS study. There were no exclusions in each experimental group. All mice were sacrificed by cervical dislocation after the experiment. The animal and the protocol used in this study was approved by the Animal Ethics Committee of Tongren Hospital in China (approval no. 2022-045-01).

Western blot analysis

Western blot was performed as we previously reported (Zhao et al., 2023). Briefly, total protein was extracted from the corpus callosum of brain tissues by radioimmunoprecipitation assay lysis (RIPA) (Beyotime, Jiangsu, China) buffer containing phenylmethyl sulfonylfluoride (PMSF) (Beyotime, Jiangsu, China) on ice for 30 min. Then, the protein was centrifuged at 4 °C at 12,000 g for 10 min and the supernatant was collected. SDS-PAGE protein buffer (Beyotime, Jiangsu, China) added to the total protein and boiling at 100 °C for 5 min. A total of 12% SmartPAGETM Precast Protein Gel (Smart Lifescience, Jiangsu, China) were used for sodium dodecyl sulfate polyacrylamide gel electrophoresis (SDS-PAGE) following the manufacturer’s protocol. About 10 μg protein of each sample were loaded in each lane of western blot. Antibodies of MAG (9043S, 1:1,000; Cell Signaling Technology (CST), Danvers, MA, USA), MOG (45268S, 1:1,000; CST, Danvers, MA, USA), MBP (78896S, 1:1,000; CST, Danvers, MA, USA), GFAP (80788S, 1:1,000; CST, Danvers, MA, USA), β-Actin (81115-1-RR, 1:1,000; Proteintech, Rosemont, IL, USA) were used for detected the relative protein expression. The relative protein expression was normalized to β-Actin.

Immunofluorescence

After the animals were routinely anesthetized and the hearts perfused, mouse brains were harvested and fixed with 4% PFA, followed by the corpus callosum region. IF were performed as reported previously (Ai et al., 2022). For immunofluorescence assays, paraffin section (4 μm) of brain tissues were performed for IF. The slices after baking, dewaxing, and gradient alcohol rehydration are first heated in 1X EDTA antigen retrieval solution to detect the minimum value of the antigen. Then, the section was rinsed with phosphate buffered saline (PBS) at room temperature, soaked in TritonX-100 0.5% liquid and put it in PBS for 15 min for further permeation. After inactivating the endogenous peroxidase with 3% H2O2 for 10 min, the sections were washed three times with PBS, blocked with 5% BSA for 30 min, incubated with the primary antibody overnight at 4 °C, washed three times with PBS at room temperature, and the second antibody was added and incubated at 37 °C for 50 min and rinse with PBS three times. Finally, after DAPI was counter-stained for 10 min, rinsed with PBS three times, dropwise anti-fluorescence quencher was added, and the slides were mounted before microscopy. Images were captured by Nikon Eclipse E600 inverted fluorescent microscope.

Antibodies of MAG (9043S, 1:100; Cell Signaling Technology (CST), Danvers, MA, USA), MOG (45268S, 1:200; CST, Danvers, MA, USA), MBP (78896S, 1:100; CST, Danvers, MA, USA), GFAP (80788S, 1:50; CST, Danvers, MA, USA) were used for IF.

Luxol fast blue (LFB) staining

Paraffin sections (4 μm) deparaffinized and incubated in LFB solution (ServiceBio, Wuhan, Hubei, China) for 4 h at 60 °C. After cooling for 15 min, the slides were washed with running water for 1 min, and soak in 70% water soak in ethanol for 10 s. Then, the sections were rinsed again and immersed in lithium carbonate solution for 1 min. This step was repeated three times until the background color fades. Finally, the slices are dehydrated and covered with neutral balsam. These images were observed by an optical microscope (Nikon, Minato City, Tokyo, Japan).

Samples preparation for LC-MS

The corpus callosum of brain tissues samples were ground into powder by adding liquid nitrogen, adding TRIS-HCl according to the volume ratio of 1:3, heating and boiling for 10 min, and then crushed by ultrasound at 100 HZ in ice water bath, over 5 s, at an interval of 5 s, ultrasonic for 2 min. Then, the final concentration of 1 M ice ethyl acid was added into the sample tube, and vortex for 2 min. Add the final concentration of about 50% acetonitrile. Centrifuge at 12,000g at 4 °C for 10 min, remove the supernatant and transfer it to a clean EP tube for freeze-drying. Add 80% acetone solution, vortex, shake, water bath ultrasound for 2 min, then, 20,000g centrifuge for 30 min at 4 °C, take the supernatant, transfer to a clean EP tube, freeze drying. Redissolved with 200 μL 0.1% TFA solution, desalt with 80 μg C18. Freeze-dried for LC-MS.

Liquid chromatography-mass spectrometry (LC-MS) analysis

Peptide analysis was performed with the Q Exactive HF (Thermo Fisher Scientific, Waltham, MA, USA) as previously reported (Lyratzakis et al., 2021). For LC-MS/MS injection, 2.5 μg dried peptide fractions were dissolved in 2% acetonitrile with 0.1% formic acid, and subsequently loaded on reverse phase columns (trapping cartridge 5 µm C18-beads, L = 5 mm, inner diameter = 100 µm; Thermo Fisher Scientific, Waltham, MA, USA), analytical column (C18, 3 µm, 150 mm × 75 µm; Eksigent, Dublin, CA, USA). Eluted peptides were separated over a 78 min gradient of water (buffer A: water with 0.1% formic acid) and acetonitrile (buffer B: acetonitrile with 0.1% formic acid). Typically, gradients were ramped from 5% to 32% B in 70 min at flowrates of 600 nL min−1. Electrospray voltage of 2.0 kV vs. the inlet of the mass spectrometer was used. The mass spectrometer was operated in data-dependent acquisition mode to automatically switch between Orbitrap-MS and ion trap acquisition. Survey of full-scan MS spectra (from m/z 300 to 1,400) were acquired in the Orbitrap with a resolution of R = 120,000. Target ions already selected for MS/MS were dynamically excluded for 18 s and the minimum intensity was 5,000.

Bioinformatics analysis

Bioinformatics analysis were performed as previously reported (Dong et al., 2021). Briefly, UniProt (http://beta.uniprot.org, Release 2021_03) were used for identified the peptide precursor protein. Isoelectric point (pI) and Mw of DEPs were calculated by the ProParm tool (https://web.expasy.org/protparam/). GO and KEGG pathways analysis of DEPs precursor protein was predicted by websites (http://geneontology.org/ and http://www.kegg.jp/). STRING (https://string-db.org) were used for predicted the interaction of DEPs precursor protein. Interactions was set up based on the experiments, databases, co-expression, neighborhood, gene fusion and co-occurrence. The heatmap and volcano plot of DEPs were generated by Version 3.2.2 of R software. The predict bioactive of DEPs were predicted by PeptideRanker (http://distilldeep.ucd.ie/PeptideRanker/).

Statistical analysis

Data were analyzed by the GraphPad Prism with unpaired t-test. A p < 0.05 was indicated as a statistically significant difference.

Results

Establishment of cuprizone-induced demyelination of central nervous system (CNS) mouse model

To investigate whether the endogenous peptides participate in the demyelination of CNS, we established a mouse model of demyelination with fed a diet containing 0.2% CPZ with normal rodent chow for 8 weeks (Fig. 1A). The body weight of the mice was measured weekly during the CPZ-fed period. During the first week of CPZ treated, there was no significant difference in body weight between CPZ-fed mice and control (20.5 ± 0.57 g vs. 21.1 ± 1.3, Fig. 1B). However, the weight growth of CPZ-induced mice was significantly lower than that of control group after 2 weeks fed (Fig. 1B). The brain tissue of corpus callosum was selected as a representative region of white matter to perform IF, Luxol fast blue (LFB) staining and western blot because it has been extensively examined in CPZ-induced animal model (Morell et al., 1998; Han et al., 2020a). IF analysis revealed that a strong loss of myelin marker MBP, MOG (Myelin oligodendrocyte glycoprotein) and MAG expression were detected in CPZ treatment compared with control (Fig. 1C), and a strong increase of glial fibrillary acidic protein (GFAP) positive area in the CPZ group compared with control group (Fig. 1C). In addition, LFB staining of brain sections showed that the well characteristic rows of interfascicular oligodendrocytes in CPZ diet group was significantly decreased than that in normal diet group (Fig. 1D). Similarly, the expression of MBP, MOG and MAG were decreased in the CC of MS mice while the expression of GFAP was dramatically increased in the CPZ-induced mice (Figs. 1E and 1F). Taken together, these results suggest that 8 weeks of CPZ-diet induced the demyelination of CNS.

Figure 1 Characterize the cuprizone induced demyelination of central nervous system in mice.

Schematic illustration of the experimental protocols, the mice fed with normal diet and 0.2% cuprizone administration diet in control group and cpz-induced group for 8 weeks, respectively (A). The body weight of the mice was measured weekly according to the experimental requirement (B). IF staining of MBP, MAG, MOG and GFAP in the corpus callosum of normal fed mice compared with cuprizone fed group (C). Luxol fast blue (LFB) staining of the CPZ-fed and control groups in the corpus callosum region (D). Western blot analysis of the protein expression of MBP, MAG, MOG and GFAP in the control group and cpz-induced group of corpus callosum (E). Statistical analysis of MBP, MAG, MOG and GFAP protein expression level by grey value analysis (F). *p < 0.05, **p < 0.01, scale bar 50 μm.

Diverse expression patterns of peptides in cuprizone-induced demyelination

To evaluate the peptide profile of CPZ-induced demyelination, corpus callosum of mouse brain was isolated, and peptides were detected by LC-MS technology. A total of 6,796 peptides were identified and 6,729 peptides were quantified. Furthermore, the expression of 217 peptides was significantly changed (fold change ≥ 2, p ≤ 0.05) in CPZ-induced mice compared with controls, which including 181 up-regulated peptides and 36 down-regulated peptides (Figs. 2A and 2B). The upregulated peptides (fold change ≥ 5) and downregulated peptides (Fold Change ≤ 0.04) in CPZ-treatment mice compared with the control group are shown in Table 1, respectively.

Figure 2 Hierarchical clustering and volcano plot of differentially expressed peptides.

Each row represents a differentially changed polypeptide, and the polypeptide sequence is listed on the right side of each row (A). Volcano plot of differentially expressed peptides (B).

Table 1 Differentially expressed peptides in CPZ-treated mice brain compared with control (fold change >5 or <0.4).

Peptide	Length	Precursor protein	FC	p value	
VGSKTKEGVVHGVTTV	16	SYUA	15.1120	0.0324	
LENPKKYIPGTK	12	CYC	8.1408	0.0144	
PHLRIRTKPFPWGDGNHTL	19	CX6A1	7.9235	0.0287	
AASVDLELKKAFTELQAKV	19	PFD1	7.1294	0.0343	
AAQKINEGLEHLAKAEKYLKTGFL	24	SNAG	6.9400	0.0366	
AWGKIGGHGAEYGAEALE	18	HBA	6.9020	0.0191	
IEGRAPVISGVTKA	14	TPPP	6.7492	0.0264	
TNPEHASDAMRAMNGESL	18	RBM3	6.6180	0.0432	
RLLLPGELAKHAVSEGTKAVTK	22	H2B1C	6.1869	0.0206	
VEVMPQNQKAIGNALKSWNET	21	ATP5H	6.0276	0.0215	
YFNKPDIDAWELRKGMNTL	19	COX5A	5.9160	0.0011	
TLPTKETIEQEKR	13	TYB10	5.8981	0.0420	
AGRKLALKTIDWVSFVEV	18	ATP5H	5.8031	0.0002	
ASATRVIQKLRNWASGQDLQAK	22	NDUA7	5.7645	0.0134	
ASQSQGIQQLLQAEKRAAEKV	21	VATG1	5.6718	0.0170	
VATLGVEVHPL	11	RAN	5.5528	0.0151	
IYEKPQTEAPQVTGPIEVPVVRT	23	CRIP2	5.5323	0.0144	
VMPQNQKAIGNALKSWNETFHARLA	25	ATP5H	5.3956	0.0287	
SVPAGRRSPTSSPTPQR	17	DYN1	5.3146	0.0333	
SIPSTANRPNRPK	13	KCMA1	5.2150	0.0271	
MESVKQRILAPGKEGIKNFAG	21	VGLU2	5.1889	0.0354	
MEPSLATGGSETTRLVSARDR	21	ZNT3	5.1865	0.0255	
ELRPTLNEL	9	COX5A	5.1286	0.0159	
KPFPWGDGNHT	11	CX6A1	5.1087	0.0113	
SLGGGTGSGMGTLLISKI	18	TBB4B	5.1065	0.0156	
YPQSKWQEQ	9	SCG1	0.3647	0.0444	
LADPTGAFGKATDL	14	PRDX5	0.3607	0.0224	
RATSNVFAMFDQSQIQEF	18	MYL9	0.3497	0.0144	
ASLKPEFVDIINAKQ	15	TPIS	0.3455	0.0392	
TGILDSIGRFFSGDRG	16	MBP	0.3355	0.0056	
ELFADKVPKTAENF	14	PPIA	0.3321	0.0012	
RTLAGNPKATPPQIVNGNH	19	SH3L3	0.3170	0.0401	
FGGFTGARKSARKLANQ	17	PNOC	0.2944	0.0331	
AYGFRDPGPQL	11	CMGA	0.2898	0.0131	
GFLVGGASLKPEFVDIINAKQ	21	TPIS	0.2876	0.0337	
APLVETSTPL	10	PCS1N	0.2780	0.0305	
ASLDKFLASVSTVL	14	HBA	0.2643	0.0126	
ADPTGAFGKATDL	13	PRDX5	0.2613	0.0289	
AGQAFRKFLPLFDRVLV	17	CH10	0.2607	0.0263	
RKFFVGGNWKMNGRKK	16	TPIS	0.2471	0.0434	
KGNDISSGTVLSDYV	15	PEBP1	0.2456	0.0294	
TQVVHESFQGRSR	13	MAG	0.2358	0.0148	
SPFYLRPPSFLRAPSWIDTGL	21	CRYAB	0.2281	0.0138	
APSGRMSVLKNLQSLDPSHRISD	23	CCKN	0.1845	0.0386	
RLLVVYPWTQR	11	HBB1	0.1146	0.0162	
KGFGYAEFEDLDSLLSAL	18	IF4B	0.1044	0.0005	
KMIYASSKDAIK	12	COF1	0.1038	0.0142	
VPRGEAAGAVQELA	14	PCS1N	0.1021	0.0017	
NDRTIEGDFLWSDGAPLLYE	20	PGCB	0.1017	0.00004	
DYAGVTVDELGKVLTPTQVM	20	PEBP1	0.1012	0.0226	
KVAFSAVRSTN	11	CBLN4	0.0981	0.0023	
TQAGSEVSALLGRIPSAVGY	20	ATPB	0.0972	0.0017	
MDQLAKELTAE	11	CMGA	0.0768	0.0099	
AHLLEAERQE	10	PCS1N	0.0729	0.0169	

Characteristics of differentially expressed peptide

The physical and biochemistry characteristics of 217 DEPs were analyzed. The results showed that the molecular weight (Mw) of the major up-regulated DEPs in CPZ group was concentrated in the range of 1,000 to 3,000 D (Fig. 3A). However, the molecular weights of the down-regulated peptides were mostly concentrated in the range of 1,400 to 2,600 D (Fig. 3A). The isoelectric point of DEPs was mostly distributed in the range of 3 to 12 (Fig. 3B). Furthermore, the Mw vs. pI of DEPs was distributed within the area of 2 to 6 and 8 to 12 (Fig. 3C). Endogenous peptides were usually produced by protein precursors, we identified 35 proteins that generated more than two peptides. In particular, 11 differentially expression proteins were produced by MBP (Fig. 3D) (Table 2). According to the analysis of N-terminal and C-terminal amino acid of the up-regulated DEPs, it was determined that alanine (A), arginine (R), leucine (L), lysine (K), serine (S) and valine (V) appeared more frequently (Fig. 3E). Similarly, alanine (A), arginine (R), leucine (L), lysine (K) and serine (S) were also the amino acid with high frequently in the N-terminal and C-terminal amino acid of down-regulated DEPs (Fig. 3F).

Figure 3 Characteristics and cleavage patterns of DEPs.

Distribution of the molecular weights of DEPs (A). Distribution of the isoelectric points of DEPs (B). Distribution of isoelectric point vs. molecular weight in DEPs (C). Peptide numbers for each protein precursor of DEPs (D). Features of the cleavage sites in upregulated DEPs (E). Features of the cleavage sites in downregulated DEPs (F).

Table 2 Differentially expressed peptides that may be involved in the myelin dysfunction of OLs.

Peptide	Location	Functional domain	Predicted
bioactivity	Precursor protein	FC	Homology	
ILDSIGRFFSGDRGAPKRGSG	169–189	Myelin_HBP	0.33	MBP	4.38	N	
RHGFLPRHRDTGILDSIGR	157–175	Myelin_HBP	0.55	MBP	4.32	Y	
ATASTMDHARHGF	148–160	Myelin_HBP	0.49	MBP	4.06	Y	
RHRDTGILDSIGRFF	163–177	Myelin_HBP	0.45	MBP	3.09	Y	
RTQDENPVVHFFKNIV	210–225	Myelin_HBP	0.40	MBP	2.81	Y	
ASTMDHARHGFLPRH	150–164	Myelin_HBP	0.34	MBP	2.74	Y	
STMDHARHGFLPRHRDTGILDSIGR	151–175	Myelin_HBP	0.33	MBP	2.70	Y	
HRDTGILDSIGRFFSG	164–179	Myelin_HBP	0.31	MBP	2.19	N	
RDTGILDSIGRFFSGDR	165–181	Myelin_HBP	0.30	MBP	2.01	N	
DTGILDSIGRFFS	166–178	Myelin_HBP	0.21	MBP	0.48	N	
TGILDSIGRFFSGDRG	167–182	Myelin_HBP	0.19	MBP	0.33	N	
TQVVHESFQGRSR	78–90	IgV_CD33	0.13	MAG	0.23	Y	
Note:

MBP, function domain 147–248, all identified peptide derived from MBP were all location in Myelin HBP function domain.

GO and KEGG pathway analysis

To gain overall insights into the function differences between these differentially expressed peptides, we preformed GO and KEGG pathways analysis using precursor proteins of all 217 discrepancy expressed peptides. Systematic analysis of cellular component, molecular function and biological process showed that myelin sheath, microtubule binding and transport are the most prominent aspects that are correlated with changes in CNS function, respectively (Figs. 4A–4C). KEGG analysis uncovered the distinct expressed peptides related to oxidative phosphorylation, Alzheimer’s disease and Parkinson’s disease (Fig. 4D).

Figure 4 GO and KEGG pathway analyses of precursor proteins of DEPs.

The top 10 enrichment biological process (BP) categories (A). The top ten enrichment cellular component (CC) categories (B). The top 10 enrichment molecular function (MF) categories (C). The top 10 enrichment pathway analysis (D).

Given that myelin related protein is crucial in maintaining OLs states, and the DEPs function enriched in myelin sheath, we next analyzed myelin-related protein derived peptides expressed levels in CPZ-induced mice compared with control. A total of 11 peptides derived from MAB and one peptide derived from MAG were differently expressed in the CC of CPZ-treated mice (Fig. 5). As shown in Table 2, DEPs of MAB and MAG were localized at Myelin_HBP and IgV_CD33 functional domain, respectively. The sequences of peptide excreted from MBP157–175, MBP148–160, MBP163–177, MBP110–125, MBP150–164, MBP151–175 and MAG78–90 were homologous with human MBP or MAG protein, respectively. Furthermore, PeptideRanker was used for predicted the bioactivity of these peptides, among 13 DEPs, the predicted bioactivity of MBP157–175 peptide (RHGFLPRHRDTGILDSIGR) were more than 0.5, indicated that MBP157–175 peptide may participate in the mechanisms of disease progression.

Figure 5 Relative intensity of 11 peptides derived from MBP and one peptide derived from MAG in the CC of CPZ-treated mice.

*p < 0.01, **p < 0.05.

Interaction network analysis

MS is mainly caused by demyelination injury of CNS. GO analysis of DEPs precursor protein revealed that 34 proteins were involved in myelin sheath process. To uncover the potential interaction between these DEPs which involved in myelin biological processes, STRING was used to conduct association network analysis of the precursor proteins of these DEPs. The results shown that these 34 proteins had interaction with each other (Fig. 6A). In addition, we also found that these 34 proteins paly a core role in the whole DEPs precursor protein interaction (Fig. 6B). Taken together, these results suggest that those DEPs involved in the progress of myelin sheath may play an important role in MS.

Figure 6 Interaction network analysis of precursor proteins of differentially expressed peptides according to STRING.

The interaction network analysis of precursor proteins of DEPs involved in myelin sheath progress (A). The interaction network analysis of precursor proteins of all DEPs (B).

Discussion

Multiple sclerosis is a highly heterogeneous disease with different pathologies and clinical manifestations. Until now, the pathogenesis of MS is still poorly understood. Moreover, there is no definite therapy for MS (Correale et al., 2017). In the present study, we investigated the profile of differentially expressed peptides in a cuprizone-induced demyelination mouse model of multiple sclerosis. We demonstrated that the identified peptides have specific physical and chemical properties due to each peptide have different molecular weight and PI. This study was the first to determine potential functional peptides that may be participate in the pathophysiology of MS and may be serve as novel diagnostic biomarkers for MS.

It has been previously shown that Cuprizone could inhibit the mitochondrial function, thereby induce oxidative stress in the corpus callosum (Shiri et al., 2021). We found that the ATP synthesis coupled proton transport, ATP biosynthetic process and ATP metabolic process were perturbed in MS mouse model in BP analysis. Besides, CC analysis suggested that the mitochondrial inner membrane, mitochondrion and mitochondrial proton-transporting ATP synthase complex were destroyed in CPZ-induced MS model. Moreover, MF analysis demonstrated that the proton-transporting ATP synthase activity and ATPase activity were dramatically affected. Furthermore, the oxidative phosphorylation was the most enrichment pathway in the KEGG analysis of the precursor protein of dysregulated peptides in CPZ-induced MS model, which is consistent with previous reports. Notably, the KEGG analysis also reveal that the precursor protein of dysregulated peptides involved in degenerative disease, including Alzheimer’s disease and Parkinson’s disease, suggesting dysregulated peptides may play an important role in degenerative diseases.

In particular, the present study confirmed that the DEPs might participate in the myelin sheath which accordance with the pathogenesis of MS. The myelin sheath is composed of multiple lipids and surrounding the axons of neurons in both peripheral nervous system and CNS (Min et al., 2009). The function of myelin sheath servers as an electrical insulator, which make nerve impulse conduction faster and more efficient than that unmyelinated nerves (Webster, 1898). We found that dysregulated peptides from 34 proteins were involved in myelin progression in CPZ-treatment MS mouse. The protein-protein interaction network analysis revealed that these 34 proteins interact with each other. The result could accelerate the understanding of the biological processes and molecular crosstalk underlying MS pathogenesis.

Peptide has the advantages of small size, can penetrate the cell membrane, easy synthesis, high activity, good efficacy, good tolerance (Del Gatto, Saviano & Zaccaro, 2021). As a result, more and more peptides are entering clinical trials and being approved as drugs. Accumulated evidence demonstrates that peptides play an important role in modulating neuronal activity, which including neuropeptide Y and amylin (McGonigle, 2012; Fosgerau & Hoffmann, 2015). According to the structure, peptides can be divided into liner or cyclic peptides (Yuan, 2020). It has been reported that the immunotherapy is a potential useful method for treatment of MS. For example, PLP139–151 mutant peptides, MBP83–99 and MBP87–99 mutant peptides benefit for disease progression by dysregulated immune cells (Katsara et al., 2008, 2009, 2014). In addition, the myelin-based autoantigen peptides, MBP30–44, MBP130–144, MBP140–154 and MBP83–99, was able to alleviate the clinical disability, to decrease the T-cell and B-cell infiltration in the spinal cord (Del Gatto, Saviano & Zaccaro, 2021). In our study, a total of 11 dysregulated peptides which derived from MBP were identified in corpus callosum of MS mouse model, and all of them were belong to the Myelin_HBP functional domain. In addition, six of the 11 MBP-derived peptides were homologous to the peptide segment of human MBP, which including MBP157–175, MBP148–160, MBP163–177, MBP110–125, MBP150–164 and MBP151–17. Interestingly, one of the MBP peptides MBP157–175 had a predicted bioactivity of 0.55, indicating that it might be a functional peptide. However, whether these DEPs participate in the pathogenesis or recovery of MS need to be further investigated.

Conclusions

In conclusion, the most novel and important finding in this study was identified the profile of DEPs in the CC of CPZ-induced mice compared to controls. The potential relationship between peptides and MS has been studied for the fist time in the brain tissue of mouse model. In addition, GO and KEGG analysis revealed that these DEPs precursor proteins play essential role in the pathogenesis of MS. Furthermore, interaction network analysis of these DEPs precursor proteins showed the connection between the DEPs. Exploring the peptidomic in the brain of MS mice may provide powerful resources for future investigations of the mechanisms of demyelination of MS and its pathogenesis, which would facilitate the discovery of a novel and effective strategy for MS treatment.

Supplemental Information

Supplemental Information 1 Graphical abstract.

Click here for additional data file.

Supplemental Information 2 GO and KEGG.

Click here for additional data file.

Supplemental Information 3 Supplemental information.

Click here for additional data file.

Supplemental Information 4 The ARRIVE guidelines 2.0: author checklist.

Click here for additional data file.

Abbreviation list

CNS Central Nervous System

MS Multiple Sclerosis

PLP Proteolipid Protein

MBP Myelin Basic Protein

MOG Myelin Oligodendrocyte Glycoprotein

FC Fold Change

CC Corpus Callosum

GO Gene Ontology

BP Biological Process

CC Cellular Component

MF Molecular Function

KEGG Kyoto Encyclopedia of Genes and Genomes

DEPs Differently Expressed Peptides

MAG Myelin-associated Glycoprotein

OLs Oligodendrocytes

CPZ bis-cyclohexanone-oxaldihydrazone

CC Corpus Callosum

RIPA Radioimmunoprecipitation Assay Lysis

PMSF Phenylmethyl Sulfonylfluoride

PBS Phosphate Buffered Saline

LC-MS Liquid Chromatography-Mass Spectrometry

LFB Luxol Fast Blue

GFAP Glial Fibrillary Acidic Protein

Mw Molecular Weight

Additional Information and Declarations

Competing Interests

Author Contributions

Animal Ethics

Peptide Deposition

Data Availability

The authors declare that they have no competing interests.

Xiaohua Dong performed the experiments, prepared figures and/or tables, and approved the final draft.

Shuchen Sun performed the experiments, prepared figures and/or tables, and approved the final draft.

Jie Li analyzed the data, prepared figures and/or tables, and approved the final draft.

Sen Shen analyzed the data, authored or reviewed drafts of the article, and approved the final draft.

Wanting Chen analyzed the data, authored or reviewed drafts of the article, and approved the final draft.

Tongqi Li conceived and designed the experiments, authored or reviewed drafts of the article, and approved the final draft.

Xinyuan Li conceived and designed the experiments, authored or reviewed drafts of the article, and approved the final draft.

The following information was supplied relating to ethical approvals (i.e., approving body and any reference numbers):

The animal used in this study was approved by the Animal Ethics Committee of Tongren Hospital in China (approval no. 2022-045-01).

The mass spectrometry proteomics data have been deposited to the ProteomeXchange Consortium (http://proteomecentral.proteomexchange.org) via the iProX partner repository (Ma et al., 2019, Chen et al., 2022).

The mass spectrometry proteomics data are available at the ProteomeXchange Consortium: PXD043468.

The following information was supplied regarding data availability:

Raw data is available in the Supplemental Files.

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
