# Peer review of "Identification of potential functional peptides involved in demyelinating injury in the central nervous system"

_PeerJ, doi:10.7717/peerj.15846_

## Round 0.1 · original submission · Minor Revisions

Please address the concerns of all reviewers and amend the manuscript accordingly.

Reviewer 1 ·

Basic reporting

-

Experimental design

1. Please add ref. For bioinformatics analysis and western blot sections.
2. Add abbreviation list.
3. Please more explain discussion section.
4. Key words?
5. Running Title?

Validity of the findings

-

Additional comments

-

·

Basic reporting

Summary:
In this study, the author identifies endogenous peptides that associate with Cuprizone-induced Multiple Sclerosis (MS) mouse model. The author first showed that Cuprizone successfully induced demyelination in the mouse model. After collecting Corpus Callosum for LC-MSMS peptidomics analysis, the author showed that 217 peptides are differentially expressed in Cuprizone-induced mouse tissue. Most of those 217 peptides are associated with myelin sheath or oxidative phosphorylation.

Overview and General Recommendation
Multiple Sclerosis (MS) is an autoimmune-mediated demyelinating disease of the central nervous system that affects young adults. MS is histologically characterized by widespread loss of myelin sheath in the brain and spinal cords. MS is clinically characterized by progressive neurological deterioration over time. Though the clinical diagnosis of MS is well established, the cause of MS is unknown, and more importantly, there is no cure for the disease.
One approach to slow the progression of MS is to use exogenous peptides to modulate the inflammatory process of MS. However, there is currently no good strategy to identify such peptides as candidates for MS treatment.
In this study, the author identifies peptides differentially expressed in the Cuprizone-induced MS mouse model and suggests that these endogenous peptides could serve as a good starting point to investigate the therapeutic role of peptides in MS.
The manuscript is generally fairly well-written and can be easily understood. The results are simple to understand, and the quality of the figures is good. The study will provide helpful information for those who want to explore peptides as a therapeutic modality for MS in the future. The major area for improvement of this manuscript is in arrangement and writing, as mentioned below. Though the study lacks a unifying story or immediate application of the findings, the results are valid, and I recommend this study for publication with minor revision.

Major Comments
1. In general, the manuscript could benefit from English language editing so that the audience could understand the message the authors want to convey more clearly.
2. In the introduction section
a. In general, the writing of the section needs solid and convincing arguments why studying peptides in MS is necessary. The author starts the section by explaining what MS is and its pathogenesis. However, in paragraph 3, there is a flow gap when the author jumps from MS description to peptides. The author has the element of supporting arguments, but it could benefit from some rearrangement, especially in paragraph 3.
b. Relating to major concern #2b, could the author provide more justification for why studying peptides in MS could be helpful in pathogenesis and therapeutics? For example, peptides are readily permeable to the blood-brain barrier and serve as a good therapeutic modality.
c. All in all, Paragraphs 3 and 4 should be rearranged. The author could focus paragraph 3 on peptides' role as a diagnostic tool in MS, and paragraph 4 focuses on the therapeutic role.

Experimental design

3. In the method section, there are some missing details that could be filled
. For Western blot analysis, how many grams of brain tissue were used, and which part of the brain tissue? Is it just the corpus callosum or the whole brain?
a. How many grams of protein were loaded in each lane of Western blot?
b. For the Immunofluorescence section, I am aware that there are multiple methods to do it. Could the author provide a reference to the method that the author decided to use? What kind of microscope did the author use?
c. In the LC-MS section, it is not clear
- How 'dried peptide fraction' was harvested?
- How much of the peptides were loaded into the reverse phase column?
- How the peptide abundance was quantified?
- I also noticed that the gradient time is longer than usual (70 mins). Could the author explain the reason for the long gradient time?
The author used STRING interaction analysis. However, the author should have mentioned how the interaction was set up. What evidence mode did the author use to generate such interactome?

Validity of the findings

4. In the result section
For Figures 1C and 1D, it would be more apparent to the audience if the author could label the structure in the IF image, such as Corpus Callosum, etc.
b. In the description of Figure 2, the author mentions hierarchical clustering. However, the author did not mention how the hierarchical clustering cutoff was set, nor did it mention hierarchical clustering in Lines 180 – 187, which I believe is related to Figure 2. Also, the author should point to which file in supplementary data the author used to generate the figure.
c. The heatmap in Figure 2 is difficult to navigate. Also, in Lines 180 – 187, the author mentions Fold Change and p-Value. I believe that a volcano plot might be more suitable for this figure.
d. In the 'Interaction network analysis' part, the author claims that 34 proteins have 'strong' interaction. However, the word 'strong' is vague. Can the author provide some metrics to describe the strength of interaction, for example, average strength of the whole interactome? Also, I am looking for the exported raw data from STRING, but I could not find one. Could the author point to me where such data is?
5. In discussion section
a. Paragraph 1, I would caution in claiming that oxidative phosphorylation DEPs is associated with MS, because the author uses CPZ, which is a known cytochrome inhibitor. The signature might arise from CPZ treatment itself and not MS-related.
b. In Paragraph 3, can the author reference why peptides are considered better candidates for treatment?
c. In paragraph 3, the mention of neuropeptides is out of context. Can the author provide some transition from the first and second sentences in paragraph 3?
d. In my opinion, paragraph 3 of discussion section should be moved to introduction where the author can demonstrate the point that why peptides in MS worth studying.

Additional comments

Minor Comments
1. Please reformat all in-line citations to (author, year) format.
2. Line 54: Can the author provide a reference
3. Line 60: I suggest taking the parenthesis out and rewriting as 'Myelin basic protein (MBP) and myelin-associated glycoprotein (MAG) are important for oligodendrocyte-mediated myelin sheath maturation.
4. Lines 63-64: Is the author suggesting that peptides involve in the re-myelination process of MS?
5. Line 65: Consider rewriting to 'Peptides are fragments of proteins that produced by …..'
6. Line 66: Consider rewriting to 'Multiple evidences suggest that peptides….. . For example, ……'
7. Lines 78 – 83: should be a separate paragraph.
8. Lines 95 – 99: I believe the author wants to say that of 6 mice in each treatment group. 3 were sacrificed for histological and IF study, while the other 3 were sacrificed for LC-MS study. I think it should be rewritten to make it clearer.
9. Lines 99 – 100: Consider remove 'In our animal experiment, no criteria were set'. I am not so sure what 'criteria' means in this case.
10. Lines 112 -113: missing the antibodies for b-Actin
11. Line 122: PBS after -> PBS at
12. Line 122: 'two drops' of what?
13. Line 165: Consider changing 'After 1 week' to 'During the first week.'
14. Line 171: What does MOG stand for?
15. Line 184: Does FC mean Fold Change?
16. Line 190: What is the 'features'? May I suggest rewriting it to 'The physical and biochemistry characteristics of 217 DEPs were analyzed'. Also, could the author explain what 'by informatics' means?
17. Line 213: Consider changing 'expressed' to 'expression.'
18. Line 225: Change CNS, to CNS.
19. Line 226: Change progress to process
20. Line 267: Change 'international' to 'interaction'
21. Line 268: What is 'pentation.'
22. Figure 1B: missing x-axis label
23. Figure 1F: How was the relative protein expression calculated? Was it normalized to b-Actin bands? Could the author describe that in the method section?
24. Figure 2: The color bar lacks a label, is it fold change?
25. Figure 4: The title of each subpanel, i.e. BP,CC, MF. What do those abbreviations stand for? I can see it in the figure description, but could the author add the parenthesis to describe it?

Reviewer 3 ·

Basic reporting

The original article by Dong et al. "Identiûcation of potential functional peptides involved in demyelinating injury in the central nervous system" demyelinating injury in the central nervous system and multiple sclerosis. In this sense, this remains to be potentially interesting for the PeerJ readers. I regard the main point of this paper as highly attractive as well as the results are clearly presented. The text does not contain any major errors, therefore I have some minor comments and recommendations:

1. There is a need to provide slightly more expanded introduction shortly mentioning/describing pharmacoeconomical aspects of MS impact on modern healthcare.
2. The figure summarizing and clarifying the results should be added.
3. Following references should be added and properly cited within the main text
- Mela A, Poniatowski ŁA, Drop B, Furtak-Niczyporuk M, Jaroszyński J, Wrona W, Staniszewska A, Dąbrowski J, Czajka A, Jagielska B, Wojciechowska M, Niewada M. Overview and Analysis of the Cost of Drug Programs in Poland: Public Payer Expenditures and Coverage of Cancer and Non-Neoplastic Diseases Related Drug Therapies from 2015-2018 Years. Front Pharmacol. 2020 Aug 14;11:1123. doi: 10.3389/fphar.2020.01123.
- Thompson AJ, Baranzini SE, Geurts J, Hemmer B, Ciccarelli O. Multiple sclerosis. Lancet. 2018 Apr 21;391(10130):1622-1636. doi: 10.1016/S0140-6736(18)30481-1.
- Olczak M, Poniatowski ŁA, Siwińska A, Kwiatkowska M, Chutorański D, Wierzba-Bobrowicz T. Elevated serum and urine levels of progranulin (PGRN) as a predictor of microglia activation in the early phase of traumatic brain injury: a further link with the development of neurodegenerative diseases. Folia Neuropathol. 2021;59(1):81-90. doi: 10.5114/fn.2021.105137.

Experimental design

The corpus callosum dissection method should be described

Validity of the findings

Correct

Additional comments

-

---

## Round 0.2 · accepted · Accept

All concerns of the reviewers were adequately addressed and the manuscript was amended accordingly.